# Learning Invariant Feature Spaces to Transfer Skills with Reinforcement Learning

Abhishek Gupta[†,∗], Coline Devin[†∗], YuXuan Liu[†], Pieter Abbeel[†‡], Sergey Levine[†]

[†] UC Berkeley, Department of Electrical Engineering and Computer Science
[‡] OpenAI
{abhigupta,coline,svlevine}@eecs.berkeley.edu
{yuxuanliu}@berkeley.edu
{pieter}@openai.com

## Abstract

People can learn a wide range of tasks from their own experience, but can also learn from observing other creatures. This can accelerate acquisition of new skills even when the observed agent differs substantially from the learning agent in terms of morphology. In this paper, we examine how reinforcement learning algorithms can transfer knowledge between morphologically different agents (e.g., different robots). We introduce a problem formulation where two agents are tasked with learning multiple skills by sharing information. Our method uses the skills that were learned by both agents to train invariant feature spaces that can then be used to transfer other skills from one agent to another. The process of learning these invariant feature spaces can be viewed as a kind of "analogy making," or implicit learning of partial correspondences between two distinct domains. We evaluate our transfer learning algorithm in two simulated robotic manipulation skills, and illustrate that we can transfer knowledge between simulated robotic arms with different numbers of links, as well as simulated arms with different actuation mechanisms, where one robot is torque-driven while the other is tendon-driven.

## 1 Introduction

People can learn large repertoires of motor skills autonomously from their own experience. However, learning is accelerated substantially when the learner is allowed to observe another person performing the same skill. In fact, human infants learn faster when they observe adults performing a task, even when the adult performs the task differently from the child, and even when the adult performs the task incorrectly (Meltzoff, 1999). Clearly, we can accelerate our own skill learning by observing a novel behavior, even when that behavior is performed by an agent with different physical capabilities or differences in morphology. Furthermore, evidence in neuroscience suggests that the parts of the brain in monkeys that respond to the pose of the hand can quickly adapt to instead respond to the pose of the end-effector of a tool held in the hand (Umilta et al., 2008). This suggests that the brain learns an invariant feature space for the task (e.g., reaching with a tool) that is independent of the morphology of the limb performing that task. Mirror neurons also fire both when the animal performs a task and when it observes another animal performing it (Rizzolatti & Craighero, 2004; Ferrari et al., 2005). Can we enable robots and other autonomous agents to transfer knowledge from other agents with different morphologies by learning such invariant representations?

In robotics and reinforcement learning, prior works have considered building direct isomorphisms between state spaces, as discussed in Section 2. However, most of these methods require specific domain knowledge to determine how to form the mapping, or operate on simple, low-dimensional environments. For instance, Taylor et al. (2008) find a mapping between state spaces by searching through all possible pairings. Learning state-to-state isomorphisms involves an assumption that the two domains can be brought into correspondence, which may not be the case for morphologically

---

[∗]These authors contributed equally to this work.

different agents. Some aspects of the skill may not be transferable at all, in which case they must be learned from scratch, but we would like to maximize the information transferred between the agents.

In this paper, we formulate this multi-agent transfer learning problem in a setting where two agents are learning multiple skills. Using the skills that have been already acquired by both agents, each agent can construct a mapping from their states into an invariant feature space. Each agent can then transfer a new skill from the other agent by projecting the executions of that skill into the invariant space, and tracking the corresponding features through its own actions. This provides a well-shaped reward function to the learner that allows it to imitate those aspects of the "teacher" agent that are invariant to differences in their morphology, while ignoring the parts of the state that cannot be imitated. Since the mapping from the state spaces of each agent into the invariant feature space might be complex and nonlinear, we use deep neural networks to represent the mappings, and we present an algorithm that can learn these mappings from the shared previously acquired skills.

The main contributions of our work are a formulation of the multi-skill transfer problem, a definition of the common feature space, and an algorithm that can be used to learn the maximally informative feature space for transfer between two agents (e.g., two robots with different morphologies). To evaluate the efficiency of this transfer process, we use a reinforcement learning algorithm to transfer skills from one agent to another through the invariant feature space. The agents we consider may differ in state-space, action-space, and dynamics. We evaluate our transfer learning method in two simulated robotic manipulation tasks, and illustrate that we can transfer knowledge between simulated robotic arms with different numbers of links, as well as simulated arms with different actuation mechanisms, where one robot is torque-driven while the other is tendon-driven.

## 2 RELATED WORK

Transfer learning has long been recognized as an important direction in robotics and reinforcement learning (Taylor & Stone (2009)). Konidaris & Barto (2006) learned value functions on subsets of the state representation that were shared between tasks, providing a shaping reward in the target task. Taylor et al. (2007) manually construct a function to map a $Q$-function from one Markov decision process (MDP) to another. Ammar & Taylor (2012) manually define a common feature space between the states of two MDPs, and use this feature space to learn a mapping between states.

Later work by Ammar et al. (2015a) uses unsupervised manifold alignment to assign pairings between states for transfer. Like in our method, they aim to transfer skills between robots with different configurations and action spaces by guiding exploration in the target domain. The main difference from our work is that Ammar et al. (2015a) assume the presence of a feature mapping that provides distances between states, and use these (hand designed) features to assign correspondences between states in the different domains. In contrast, we assume that good correspondences in episodic tasks can be extracted through time alignment, and focus on learning the feature mapping itself. Additionally, we do not try to learn a direct mapping between state spaces but instead try to learn nonlinear embedding functions into a common feature space, as compared to linear mappings between state spaces learned in Ammar et al. (2015a). In a similar vein, Raimalwala et al. (2016) consider transfer learning across linear time-invariant (LTI) systems through simple alignment based methods. Although this method is quite effective in enabling transfer in these systems, it does not apply to the higher dimensional continuous control tasks we consider which may have non-linear dynamics, and may not be LTI.

In machine learning, Pan & Yang (2010) provide an extensive survey on transfer learning which addresses the case of train and test data being drawn from different distributions, as well as learning models that succeed on multiple, related tasks. Ben-David & Schuller (2003) derive theoretical guarantees on this sort of multitask learning and provide a formal framework for defining task relatedness. In deep learning, Caruana (1997) show that a multitask network can leverage a shared representation of the input to learn multiple tasks more quickly together than separately.

More recent work in deep learning has also looked at transferring policies by reusing policy parameters between environments (Rusu et al., 2016a;b; Braylan et al., 2015; Daftry et al., 2016), using either regularization or novel neural network architectures, though this work has not looked at transfer between agents with structural differences in state, such as different dimensionalities. Our approach is largely orthogonal to policy transfer methods, since our aim is not to directly transfer a

skill policy, which is typically impossible in the presence of substantial morphological differences, but rather to learn a shared feature space that can be used to transfer information about a skill that is shared across robots, while ignoring those aspects that are not shared. Our own recent work has looked at morphological differences in the context of multi-agent and multi-task learning (Devin et al., 2016), by reusing neural network components across agent/task combinations. In contrast to that work, which transferred components of policies, our present work aims to learn common feature spaces in situations where we have just two agents. We do not aim to transfer parts of policies themselves, but instead look at shared structure in the states visited by optimal policies, which can be viewed as a kind of analogy making across domains.

Learning feature spaces has also been studied in the domain of computer vision as a mechanism for domain adaptation and metric learning. Xing et al. (2002) finds a linear transformation of the input data to satisfy pairwise similarity contraints, while past work by Chopra et al. (2005) used Siamese networks to learn a feature space where paired images are brought close together and unpaired images are pushed apart. This enables a semantically meaningful metric space to be learned with only pairs as labels. Later work on domain adaptation by Tzeng et al. (2015) and Ganin et al. (2016) use an adversarial approach to learn an image embedding that is useful for classification and invariant to the input image's domain. We use the idea of learning a metric space from paired states, though the adversarial approach could also be used with our method as an alternative objective function in future work.

# 3 PROBLEM FORMULATION AND ASSUMPTIONS

We formalize our transfer problem in a general way by considering a source domain and a target domain, denoted $D_S$ and $D_T$, which each correspond to Markov decision processes (MDPs) $D_S = (\mathscr{S}_S, \mathscr{A}_S, T_S, R_S)$ and $D_T = (\mathscr{S}_T, \mathscr{A}_T, T_T, R_T)$, each with its own state space $\mathscr{S}$, action space $\mathscr{A}$, dynamics or transition function $T$, and reward function $R$. In general, the state and action spaces in the two domains might be completely different. Correspondingly, the dynamics $T_S$ and $T_T$ also differ, often dramatically. However, we assume that the reward functions share some structural similarity, in that the state distribution of an optimal policy in the source domain will resemble the state distribution of an optimal policy in the target domain when projected into some *common feature space*. For example, in one of our experimental tasks, $D_S$ corresponds to a robotic arm with 3 links, while $D_T$ is an arm with 4 links. While the dimensionalities of the states and action are completely different, the two arms are performing the same task, with a reward that depends on the position of the end-effector. Although this end-effector is a complex nonlinear function of the state, the reward is structurally similar for both agents.

## 3.1 COMMON FEATURE SPACES

We can formalize this common feature space assumption as following: if $\pi_S(s_S)$ denotes the state distribution of the optimal policy in $D_S$, and $\pi_T(s_T)$ denotes the state distribution of the optimal policy in $D_T$, it is possible to learn two functions, $f$ and $g$, such that $p(f(s_S)) = p(g(s_T))$ for $s_S \sim \pi_S$ and $s_T \sim \pi_T$. That is, the images of $\pi_S$ under $f$ and $\pi_T$ under $g$ correspond to the same distribution. This assumption is trivially true if we allow lossy mappings $f$ and $g$ (e.g. if $f(s_S) = g(s_T) = 0$ for all $s_S$ and $s_T$). However, the less information we lose in $f$ and $g$, the more informative the shared feature will be for the purpose of transfer. So while we might not in general be able to fully recover $\pi_T$ from the image of $\pi_S$ under $f$, we can attempt to learn $f$ and $g$ to maximize the amount of information contained in the shared space.

## 3.2 LEARNING WITH MULTIPLE SKILLS

In order to learn the common feature space, we need examples from both domains. While both agents could in principle learn a common feature space through direct exploration, in this work we instead assume that the agents have prior knowledge about each other, in the form of other skills that they have both learned. This assumption is reasonable, since many practical use-cases of transfer involve two agents that already have competence in a range of simple settings, and wish to transfer the competence of one agent in a new setting to another one. For example, we might wish to transfer a particular cooking skill from one home robot to another one, in a setting where both robots have

already learned some basic manipulation behaviors that can allow us to build a common feature space between the two robots. Humans similarly leverage their extensive prior knowledge to aid in transfer, by recognizing limbs and hands and understanding their function.

To formalize the setting where the two agents can perform multiple tasks, we divide the state space in each of the two domains into an agent-specific state $s_r$ and a task-specific state $s_{\text{env}}$. A similar partitioning of the state variables was previously discussed by Devin et al. (2016), and is closely related to the agent-space proposed by Konidaris (2006). For simplicity, we will consider a case where there are just two skills: one *proxy* skill that has been learned by both agents, and one *test* skill that has been learned by the source agent in the domain $D_S$ and is currently being transferred to the target agent in domain $D_T$. We will use $D_{Sp}$ and $D_{Tp}$ to denote the proxy task domains for the source and target agents. We assume that $D_S$ and $D_{Sp}$ (and similarly $D_T$ and $D_{Tp}$) differ only in their reward functions and task-specific states, with the agent-specific state spaces $\mathscr{S}_r$ and action spaces being the same between the proxy and test domains. For example $D_{Sp}$ might correspond to a 3-link robot pushing an object, while $D_S$ might correspond to the same robot opening a drawer, and $D_{Tp}$ and $D_T$ correspond to a completely different robot performing those tasks. Then, we can learn functions $f$ and $g$ on the robot-specific states of the proxy domains, and use them to transfer knowledge from $D_S$ to $D_T$.

The idea in this setup is that both agents will have already learned the proxy task, and we can compare how they perform this task in order to determine the common feature space. This is a natural problem setup for many robotic transfer learning problems, as well as other domains where multiple distinct agents might need to each learn a large collection of skills, exchanging their experience and learning which information they can and cannot transfer from each other. In a practical scenario, each robot might have already learned a large number of basic skills, some of which were learned by both robots. These skills are candidate proxy tasks that the robots can use to learn their shared space, which one robot can then use to transfer knowledge from the other one and more quickly learn skills that it does not yet possess.

## 3.3 ESTIMATING CORRESPONDENCES FROM PROXY SKILL

The proxy skill is useful for learning which pairs of agent-specific states correspond across both domains. We want to learn a pairing $P$, which is a list of pairs of states in both domains which are corresponding. This is then used for the contrastive loss as described in Section 4. These correspondences could be obtained through an unsupervised alignment procedure but in our method we explore two simpler approaches exploiting the fact that the skills we consider are episodic.

### 3.3.1 TIME-BASED ALIGNMENT

The first extremely simple approach we consider is to say that in such episodic skills, a reasonable approximate alignment can be obtained by assuming that the two agents will perform each task at roughly the same rate, and we can therefore simply pair the states that are visited in the same time step in the two proxy domains.

### 3.3.2 ALTERNATING OPTIMIZATION USING DYNAMIC TIME WARPING

However, this alignment is sensitive to time based alignment and may not be very robust if the agents are performing the task at somewhat different rates. In order to address this, we formulate an alternating optimization procedure to be more robust than time-based alignment. This optimization alternates between learning a common feature space using currently estimated correspondences, and re-estimating correspondences using the currently learned feature space. We make use of Dynamic Time Warping (DTW) as described in Müller (2007), a well known method for learning correspondences across sequences which may vary in speed. Dynamic time warping requires a metric space to compare elements in the sequences to compute an optimal alignment between the sequences. In this method, we initialize the weak time-based alignment described in the previous paragraph and use it to learn a common feature space. This feature space serves as a metric space for DTW to re-estimate correspondences across domains. The new correspondences are then used as pairs for learning a better feature space, and so on. This forms an Expectation-Maximization style approach which can help estimate better correspondences than naive time-alignment.

# 4 LEARNING COMMON FEATURE SPACES FOR SKILL TRANSFER

In this section, we will discuss how the shared space can be learned by means of the proxy task. We will then describe how this shared space can be used for knowledge transfer for a new task, and finally present results that evaluate transfer on a set of simulated robotic control domains.

We wish to find functions $f$ and $g$ such that, for states $s_{Sp}$ and $s_{Tp}$ along the optimal policies $\pi_{Sp}^*$ and $\pi_{Tp}^*$, $f$ and $g$ approximately satisfy $p(f(s_{Sp,r})) = p(g(s_{Tp,r}))$. If we can find the common feature space by learning $f$ and $g$, we can optimize $\pi_T$ by directly mimicking the distribution over $f(s_{Sp,r})$, where $s_{Sp,r} \sim \pi_S$.

## 4.1 LEARNING THE EMBEDDING FUNCTIONS FROM A PROXY TASK

To approximate the requirement that $p(f(s_{Sp,r})) = p(g(s_{Tp,r}))$, we assume a pairing $P$ of states in the proxy domains as described in 3.3. The pairing $P$ is a list of pairs of states $(s_{Sp}, s_{Tp})$ which are corresponding across domains. As $f$ and $g$ are parametrized as neural networks, we can optimize them using the similarity loss metric introduced by Chopra et al. (2005):

$$\mathcal{L}_{\text{sim}}(s_{Sp}, s_{Tp}; \theta_f, \theta_g) = ||f(s_{Sp,r}; \theta_f) - g(s_{Tp,r}; \theta_g)||_2.$$

Where $\theta_f$ and $\theta_g$ are the function parameters, $(s_{Sp,r}, s_{Tp,r}) \in P$.

However, as described in Section 3, if this is the only objective for learning $f$ and $g$, we can easily end up with uninformative degenerate mappings, such as the one where $f(s_{Sp,r}) = g(s_{Tp,r}) = 0$. Intuitively, a good pair of mappings $f$ and $g$ would be as close as possible to being invertible, so as to preserve as much of the information about the source domain as possible. We therefore train a second pair of decoder networks with the goal of optimizing the quality of the reconstruction of $s_{Sp,r}$ and $s_{Tp,r}$ from the shared feature space, which encourages $f$ and $g$ to preserve the maximum amount of domain-invariant information. We define decoders $\text{Dec}_S(f(s_{Sp,r}))$ and $\text{Dec}_T(g(s_{Tp,r}))$ that map from the feature space back to their respective states. Note that, compared to conven-

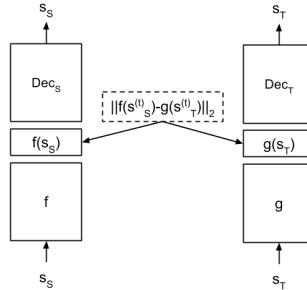

*Figure 1: The two embedding functions $f$ and $g$ are trained with a contrastive loss between the domains, along with decoders that optimize autoencoder losses.*

tional Siamese network methods, the weights between $f$ and $g$ are not tied, and in general the networks have different dimensional inputs. The objectives for these are:

$$\mathcal{L}_{\text{AE}_S}(s_{Sp,r}; \theta_f, \theta_{\text{Dec}_S}) = ||s_{Sp,r} - \text{Dec}_S(f(s_{Sp,r}; \theta_f); \theta_{\text{Dec}_S})||_2,$$

$$\mathcal{L}_{\text{AE}_T}(s_{Tp,r}; \theta_g, \theta_{\text{Dec}_T}) = ||s_{Tp,r} - \text{Dec}_T(g(s_{Tp,r}; \theta_g); \theta_{\text{Dec}_T})||_2,$$

where $\theta_{\text{Dec}_S}$ and $\theta_{\text{Dec}_T}$ are the decoder weights. We train the entire network end-to-end using backpropagation, where the full objective is

$$\min_{\theta_f, \theta_g, \theta_{\text{Dec}_S}, \theta_{\text{Dec}_T}} \sum_{(s_{Sp}, s_{Tp}) \in P} \mathcal{L}_{\text{AE}_S}(s_{Sp,r}; \theta_f, \theta_{\text{Dec}_S}) + \mathcal{L}_{\text{AE}_T}(s_{Tp,r}; \theta_g, \theta_{\text{Dec}_T}) + \mathcal{L}_{\text{sim}}(s_{Sp,r}, s_{Tp,r}; \theta_f, \theta_g)$$

A diagram of this learning approach is shown in Figure 1.

### 4.1.1 USING THE COMMON EMBEDDING FOR KNOWLEDGE TRANSFER

The functions $f$ and $g$ learned using the approach described above establish an invariant space across the two domains. However, because these functions need not be invertible, directly mapping from a state in the source domain to a state in the target domain is not feasible.

Instead of attempting direct policy transfer, we match the distributions of optimal trajectories across the domains. Given $f$ and $g$ learned from the network described in Section 4, and the distribution $\pi_S^*$ of optimal trajectories in the source domain, we can incentivize the distribution of trajectories in the target domain to be similar to the source domains under the mappings $f$ and $g$. Ideally, we would like the distributions $p(f(s_{S,r}))$ and $p(g(s_{T,r}))$ to match as closely as possible. However, it may still be necessary for the target agent to learn some aspects of the skill from scratch, since not

all intricacies will transfer in the presence of morphological differences. We therefore use a reinforcement learning algorithm to learn $\pi_T$, but with an additional term added to the reward function that provides guidance via $f(s_{S,r})$. This term has following form:

$$r_{\text{transfer}}(s_{T,r}^{(t)}) = \alpha||f(s_{S,r}^{(t)}; \theta_f) - g(s_{T,r}^{(t)}; \theta_g)||_2,$$

where $s_{S,r}^{(t)}$ is the agent-specific state along the optimal policy in the source domain at time step $t$, and $s_{T,r}^{(t)}$ is the agent-specific state along the current policy that is being learned in the target domain at time step $t$, and $\alpha$ is a weight on the transfer reward that controls its importance relative to the overall task goal. In essence, this additional reward provides a form of reward shaping, which gives additional learning guidance in the target domain. In sparse reward environments, task performance is highly dependent on directed exploration, and this additional incentive to match trajectory distributions in the embedding space provides strong guidance for task performance.

In tasks where the pairs mapping $\mathscr{P}$ is imperfect, the transfer reward may sometimes interfere with learning when the target domain policy is already very good, though it is usually very helpful in the early stages of learning. We therefore might consider gradually reducing the weight $\alpha$ as learning progresses in the target domain. We use this technique for our second experiment, which learns a policy for a tendon-driven arm.

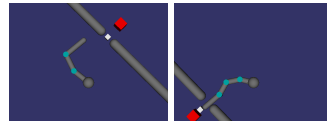

Figure 2: The 3 and 4 link robots performing the button pressing task, which we use to evaluate the performance of our transfer method. Each task is trained on multiple conditions where the objects start in different locations.

## 5 EXPERIMENTS

Our experiments aim to evaluate how well common feature space learning can transfer skills between morphologically different agents. The experiments were performed in simulation using the MuJoCo physics simulator (Todorov et al., 2012), in order to explore a variety of different robots and actuation mechanisms. The embedding functions $f$ and $g$ in our experiments are 3 layer neural networks with 60 hidden units each and ReLu non-linearities. They are trained end-to-end with standard backpropagation using the ADAM optimizer (Kingma & Ba, 2015). Videos of our experiment will be available at `https://sites.google.com/site/invariantfeaturetransfer/` For details of the reinforcement learning algorithm used, refer to Appendix A.

### 5.1 METHODS USED FOR COMPARISON

In the following experiments, we compare our method with other methods. The simplest one, referred to as "no transfer", aims to learn the target task from scratch. This method generally cannot succeed in sparse reward environments without a large number of episodes. Table 1 shows that, without transfer, the tasks are not learned even with 3-4 times more experience.

We also compare to several linear methods, including random projections, canonical correlation analysis (CCA), and unsupervised manifold alignment (UMA). Random projections of data have been found to provide meaningful dimensionality reduction (Hegde et al., 2008). We assign $f$ and $g$ be random projections into spaces of the same dimension, and transfer as described in Section 4.1.1. CCA (Hotelling, 1936) aims to find a basis for the data in which the source data and target data are maximally correlated. We use the matrices that map from state space to the learned basis as $f$ and $g$. UMA (Wang & Mahadevan (2009), Ammar et al. (2015b)) uses pairwise distances between states to align the manifolds of the two domains. These methods impose a linearity constraint on $f$ and $g$ which proves to limit the expressiveness of the embeddings. We find that using CCA to learn the embedding allows for transfer between robots, albeit without as much performance gained than if $f$ and $g$ are neural networks.

We also compare to kernel-CCA (KCCA) which uses a kernel matrix to perform CCA, allowing the method to use an implied non-linear feature mapping of the data. We test on several different kernels, including polynomial (quad), radial basis (rbf), and linear. These methods perform especially well on transfer between different actuation methods, but which kernel to use for best performance is not consistent between experiments. For example, although the quadratic kernel performs competitively with our method for the tendon experiment, it does not work at all for our button pushing experiment.

The last method we compare with is "direct mapping" which learns to directly predict $s_{T,r}$ from $s_{S,r}$ instead of mapping both into a common space. This is representative of a number of prior techniques that attempt to put source and target domains into direct correspondence such as Taylor et al. (2008). In this method, we use the same pairs as we do for our method, estimated from prior experience, but try to map directly from the source domain to the target domain. In order to guide learning using this method, we pass optimal source trajectories through the learned mapping, and then penalize the target robot for deviating from these predicted trajectories. As seen in Figures 5 and 8 this method does not succeed, probably because mapping from one state space to another is more difficult than mapping both state spaces into similar embeddings. The key difference between this method and ours is that we map both domains into a common space, which allows us to put only the common parts of the state spaces in correspondence instead of trying to map between entire states across domains.

We have also included a comparison between using time-based alignment across domains versus using a more elaborate EM-style procedure as described in 3.3.2.

## 5.2 TRANSFER BETWEEN ROBOTS WITH DIFFERENT NUMBERS OF LINKS

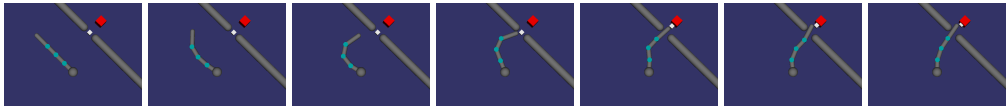

Figure 3: *The 4-link robot pushing the button. Note that the reward function only tells the agent how far the button has been depressed, and provides no information to indicate that the arm should reach for the button.*

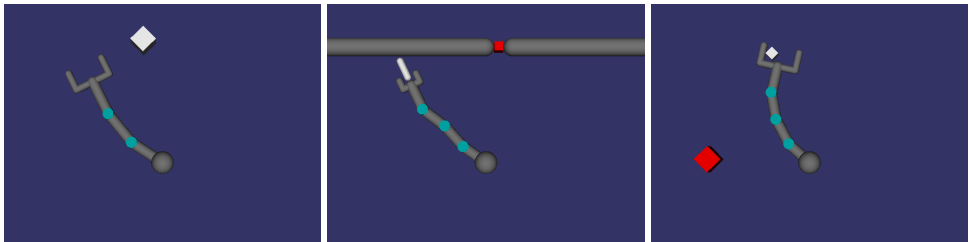

Figure 4: *The 3 and 4 link robots performing each of the three proxy tasks we consider: target reaching, peg insertion, and block moving. Our results indicate that using all three proxy tasks to learn the common feature space improves performance over any single proxy task.*

In our first experiment, we evaluate our method on transferring information from a 3-link robot to a 4-link robot. These robots have similar size but different numbers of links and actuators, making the representation needed for transfer non-trivial to learn. In order to evaluate the effectiveness of our method, we consider tasks with sparse or delayed rewards, which are difficult to learn quickly without the use of prior knowledge, large amounts of experience, or a detailed shaping function to guide exploration. For transfer between the 3 link and 4 link robots, we evaluate our method on a button pressing task as shown in Figures 2 and 3. The goal of this task is to reach through a narrow opening and press the white button to the red goal marker indicated in the figure. The caveat is that the reward signal tells the arms nothing about where the button is, but only penalizes distance between the white button and the red goal. Prior work has generally used well-shaped reward functions for tasks of this type, with terms that reward the arm for approaching the object of interest (Lillicrap et al., 2015; Devin et al., 2016). Without the presence of a directed reward shaping guiding the arm towards the button, it is very difficult for the task to be performed at all in the target domain, as seen from the performance of learning from scratch with no transfer ("baseline") in the target domain in Figure 5. This is indicative of how such a task might be learned in the real world, where it is hard to provide anything but very sparse feedback by using a sensor on the button.

For this experiment, we compare the quality of transfer when using different proxy tasks: reaching a target, moving a white block to the red goal, and inserting a peg into a slot near the robot, as shown in Figure 4. These tasks are significantly easier than the sparse reward button pressing task. Collecting successful trajectories from the proxy task, we train the functions $f$ and $g$ as described in

| Task | Button (Section 5.2) | Block Pull (Section 5.3) | Block Push (Section 5.4) |
|---|---|---|---|
| **Best in 75 iters** | 0.0% | 4.2% | 7.1% |

*Table 1: Maximum success rate of "no transfer" method over 75 iterations of training shown for the 3 tasks considered in Sections 5.2, 5.3, and 5.4. Because the target environments suffer from sparse rewards, this method is unable to learn the tasks with a tractable amount of data.*

Section 4. Note that the state in both robots is just the joint angles and joint velocities. Learning a suitable common feature space therefore requires the networks to understand how to map from joint angles to end-effectors for both robots.

We consider the 3-link robot pressing the button as the source domain and the 4-link robot pressing the button as the target domain. We allow the domain with the 3-link robot to have a well shaped cost function which has 2 terms: one for bringing the arm close to the button, and one for the distance of the button from the red goal position. The performance of our method is shown in Figure 5. The agent trained with our method performs more directed exploration and achieves an almost perfect success rate in 7 iterations. The CCA method requires about 4 times more experience to reach 60% success than our method, indicating that using deep function approximators for the functions *f* and *g* which allows for a more expressive mapping than CCA. Even with kernel CCA, the task is not able to be performed as well as our method. Additionally the UMA and random projections baselines perform much worse than our method. We additionally find that using the EM style alignment procedure described in 3.3.2 also allows us to reach perfect formance as shown in Figure 5. Investigating this method further will be the subject of future work.

Learning a direct mapping between states in both domains only provides limited transfer because this approach is forced to learn a mapping directly from one state space to the other, even though there is often no complete correspondence between two morphologically different robots. For example there may be some parts of the state which can be put in correspondence, but others which cannot. Our method of learning a common space between robots allows the embedding functions to only retain transferable information.

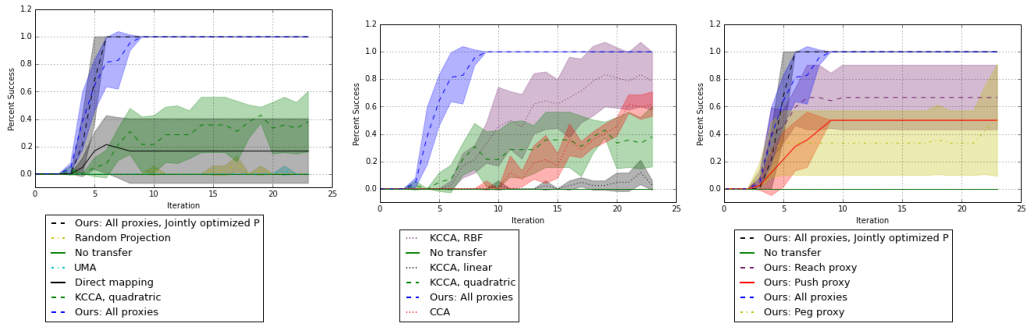

*Figure 5: Performance of 4-link arm on the sparse reward button pressing task described in Section 5.2. On the left and middle, we compare our method with the methods described in Section 5.1. On the right, the "peg," "push," and "reach" proxy ablations indicate the performance when using embedding functions learned from those proxy tasks. The embedding improves significantly when learned from all three proxy tasks, indicating that our method benefits from additional prior experience.*

## 5.3 TRANSFER BETWEEN TORQUE CONTROLLED AND TENDON CONTROLLED MANIPULATORS

In order to illustrate the ability of our method to transfer across vastly different actuation mechanisms and learn representations that are hard to specify by hand, we consider transfer between a torque driven arm and a tendon driven arm, both with 3 links. These arms are pictured in Figure 6. The torque driven arm has motors at each of its joints that directly control its motion, and the state includes joint angles and joint velocities. The tendon driven arm, illustrated in Figure 6, uses three tendons to actuate the joints. The first tendon spans both the shoulder and the elbow, while the second and third control the elbow and wrist individually. The last tendon has a variable-length

lever arm, while the first two have fixed-length lever arms, corresponding to tendons that conform to the arm as it bends. This coupled system uses tendon lengths and tendon velocities as the state representation, without direct access to joint angles or end-effector positions.

The state representations of the two robots are dramatically different, both in terms of units, dimensionality, and semantics. Therefore, learning a suitable common feature space represents a considerable challenge. In our evaluation, the torque driven arm is the source robot, and the tendon driven arm is the target robot. The task we require both robots to perform is a block pulling task indicated in Figure 7. This involves pulling a block in the direction indicated, which is nontrivial because it requires moving the arm under and around the block, which is restricted to only move in the directions indicated in Figure 6. With random exploration, the target robot is unable to perform directed exploration to get the arm to actually pull the block in the desired direction, as shown in Figure 8.

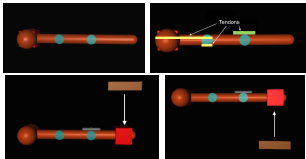

Figure 6: The top images show the source and target domain robots: the robot on the left is torque driven at the joints and the one on the right is tendon driven. The tendons are highlighted in the image; the green tendon has a variable-length lever arm, while the yellow tendons have fixed-length lever arms. Note that the first tendon couples two joints. The bottom images show two variations of the test task.

We use one proxy task in the experiment, which involves both arms reaching to various locations. With embedding functions $f$ and $g$ trained on optimal trajectories from the proxy task, we see that the transfer reward from our method enables the task to actually be performed with a tendon driven arm. The baseline of learning from scratch, which again corresponds to attempting to learn the task with the target tendon-driven arm from scratch, fails completely. The other methods of using CCA, and learning a direct mapping are able to achieve better performance than learning from scratch but learn slower. Kernel CCA with the quadratic kernel does competitively with our method but in turn performed very poorly on the button task so is not very consistent. Additionally, the random projection and UMA baselines perform quite poorly. The performance of the EM style alignment procedure is very similar to the standard time based alignment as seen in Figure 8, likely because the data is already quite time aligned across the domains. These results indicate that learning the common feature subspace can enable substantially accelerated learning in the target domain, and in fact can allow the target agent to learn a task that it fails to learn without any transfer rewards, and performs better than alternative methods.

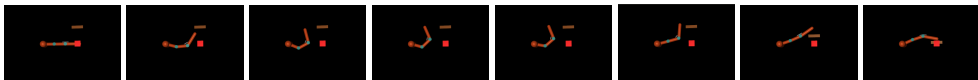

Figure 7: The tendon-driven robot pulling the block. Note that the reward function only tells the agent how far the block is from the red goal and provides no information to indicate that the arm should reach around the block in order to pull it. The block is restricted to move only towards the red goal, but the agent needs to move under and around the block to pull it.

## 5.4 TRANSFER THROUGH IMAGE FEATURES

A compelling use-case for learned common embeddings is in learning vision-based policies. In this experimental setup, we evaluate our method on learning embeddings from raw pixels instead of from robot state. Enabling transfer from extra high dimensional inputs like images would allow significantly more natural transfer across a variety of robots without restrictive assumptions about full state information.

We evaluate our method on transfer across a 3-link and a 4-link robot as in Section 5.2, but use images instead of state. Because images from the source and target domains are the same size and the same "type", we let $g = f$. We parametrize $f$ as 3 convolutional layers with 5x5 filters and no pooling. A spatial softmax (Levine et al., 2016) is applied to the output of the third layer such that $f$ outputs normalized pixel indices of feature points on the image. These "feature points" form the latent representation that we compare across domains. Intuitively the common "feature points" embeddings should represent parts of the robots which are common across different robots.

Embeddings between the domains are built using a proxy task of reaching to a point, similar to the one described in the previous experiments. The test task in this case is to push a white block

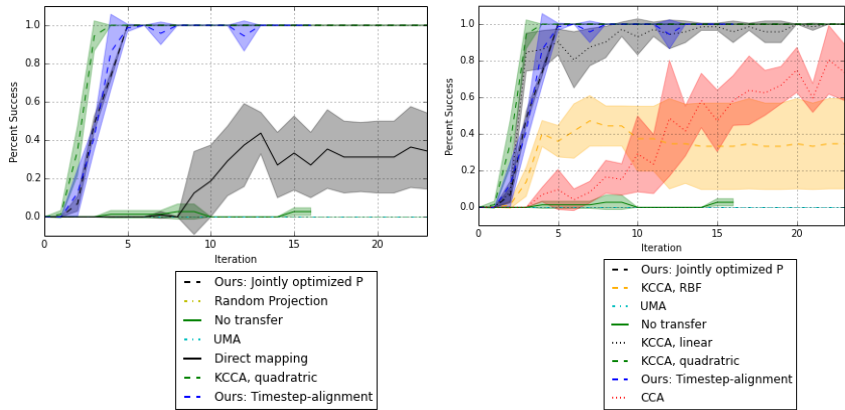

*Figure 8: Performance of tendon-controlled arm on block pulling task. While the environment's reward is too sparse to succeed in a reasonable time without transfer, using our method to match feature space state distributions enables faster learning. Using a linear embedding or mapping directly from source states to target states allows for some transfer. Optimizing over P instead of assuming time-based alignment does not hurt performance. KCCA with quadratic kernel performs very well in this experiment, but not in experiment 1.*

to a red target as shown in Figure 9a, which suffers from sparse rewards because the reward only accounts for the distance of the block from the goal. Unless the robot knows that it has to touch the block, it receives no reward and has unguided exploration. As shown in Figure 9b, our method is able to transfer meaningful information from source to target robot directly from raw images and successfully perform the task even in the presence of sparse rewards.

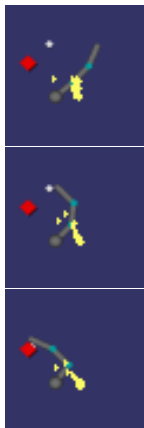

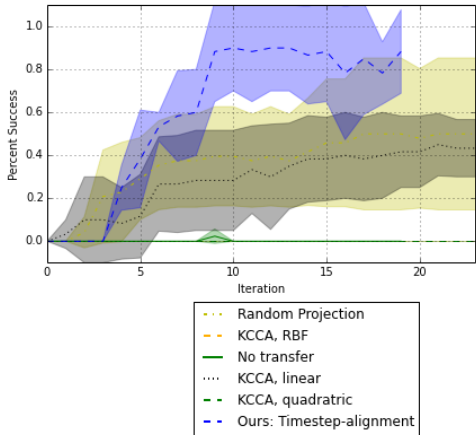

*(a) The 3-link robot demonstrating the task. The yellow triangles mark the locations of the feature points output by f applied to the image pixels. We then use the feature points to transfer the skill to the 4-link robot.*

*(b) Performance of 4-link robot on block pushing task for transfer using raw images. We transfer from the 3-link robot by learning a feature space from raw pixels of both domains, enabling effective faster learning. Random projections and linear kernel-CCA have some success in transfer. The baseline is unable to succeed because of the reward signal is too sparse without transfer.*

## 6 DISCUSSION AND FUTURE WORK

We presented a method for transferring skills between morphologically different agents using invariant feature spaces. The formulation of our transfer problem corresponds to a setting where two agents (e.g. two different robots) have each learned a collection of skills, with some skills known to just one of the agents, and some shared by both. A shared skill can be used to learn a space that implicitly brings the agents into correspondence, without assuming that an explicit state space isomorphism can be constructed. By then mapping into this space a skill that is known to only one

of the agents, the other agent can substantially accelerate its learning of this skill by transferring the shared structure. We present an algorithm for learning the shared feature spaces using a shared proxy task, and experimentally illustrate that we can use this method to transfer manipulation skills between different simulated robotic arms. Our experiments include transfer between arms with different numbers of links, as well as transfer from a torque-driven arm to a tendon-driven arm.

A promising direction for future work is to explicitly handle situations where the two (or more) agents must transfer new skills by using a large collection of prior behaviors, with different degrees of similarity between the agents. In this case, constructing a shared feature space involves not only mapping the skills into a single space, but deciding which skills should or should not be combined. For example, a wheeled robot might share manipulation strategies with a legged robot, but should not attempt to share locomotion behaviors.

In a large-scale lifelong learning domain with many agent and many skills, we could also consider using our approach to gradually construct more and more detailed common feature spaces by transferring a skill from one agent to another, using that new skill to build a better common feature space, and then using this improved feature space to transfer more skills. Automatically choosing which skills to transfer when in order to minimize the training time of an entire skill repertoire is an interesting and exciting direction for future work.

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

## 7 APPENDIX

### 7.1 REINFORCEMENT LEARNING WITH LOCAL MODELS

Although we can use any suitable reinforcement learning algorithm for learning policies, in this work, we use a simple trajectory-centric reinforcement learning method that trains time-varying linear-Gaussian policies (Levine & Abbeel, 2014). While this method produces simple policies, it is very efficient, making it well suited for robotic learning. To obtain robot trajectories for training tasks and source robots, we optimize time-varying linear-Gaussian policies through a trajectory-centric reinforcement learning algorithm that alternates between fitting local time-varying linear dynamics models, and updating the time-varying linear-Gaussian policies using the iterative linear-quadratic Gaussian regulator algorithm (iLQG) (Li & Todorov, 2004). This approach is simple and efficient, and is typically able to learn complex high-dimensional skills using just tens of trials, making it well suited for rapid transfer. The resulting time-varying linear-Gaussian policies are parametrized as $p(u_t|x_t) = \mathcal{N}(K_t x_t + k_t, C_t)$ where $K_t$, $k_t$, and $C_t$ are learned parameters. Further details of this method are presented in prior work (Levine & Abbeel, 2014).

We use the same reinforcement learning algorithm to provide solutions in the source domain $D_S$, though again any suitable reinforcement learning method (or even human demonstrations) could be used instead. To evaluate the ability of our method to provide detailed guidance through the transfer reward $r_{\text{transfer}}$, we use relatively sparse reward functions in the target domain $D_T$, as discussed below. To generate the original skills in the source domain $D_S$ and in the proxy domains $D_{Sp}$ and $D_{Tp}$, we manually designed the appropriate shaped costs to enable learning from scratch to succeed, though we note again that our method is agnostic to how the source domain and proxy domain skills are acquired.

