# Peer review of "Learning Invariant Feature Spaces to Transfer Skills with Reinforcement Learning"

_ICLR 2017 — accepted_

[Public Comment · Coline Manon Devin · 13 Dec 2016]
**Summary of Author Responses**

To address the reviewers’ questions, we have made the following changes:
-We added comparisons of our method to using canonical correlation analysis to find the embedding functions F and G (see Figures 5 and 8a)
-We added comparisons of our method to directly predicting the target states from the source states (see Figures 5, 8a)
-We have clarified the description of P in Sec 4.1
-We have added two previous approaches to the related work section.

[Official Review · AnonReviewer1 · rating 6 · confidence 4 · 16 Dec 2016]

The paper considers the problem of transferring skills between robots with different morphologies, in the context of agents that have to perform several tasks.  A core component of the proposed approach is to use a task-invariant future space, which can be shared between tasks & between agents.

Compared to previous work (Ammar et al. 2015), it seems the main contribution here is to “assume that good correspondences in episodic tasks can be extracted through time alignment” (Sec. 2).  This is an interesting hypothesis. There is also similarity to work by Raimalwala et al (2016), but the authors argue their method is better equipped to handle non-linear dynamics. These are two interesting hypotheses, however I don’t see that they have been verified in the presented empirical results.  In particular, the question of the pairing correspondence seems crucial. What happens when the time alignment is not suitable. Is it possible to use dynamic time warping (or similar method) to achieve reasonable results?  Robustness to misspecification of the pairing correspondence P seems a major concern.

In general, more comparison to other transfer methods, including those listed in Sec.2, would be very valuable.  The addition of Sec.5.1 is definitely a right step in this direction, but represents a small portion of the recent work on transfer learning.  I appreciate that other methods transfer other pieces of information (e.g. the policy), but still if the end goal is better performance, what is worth transferring (in addition to how to do the transfer) should be a reasonable question to explore.

Overall, the paper tackles an important problem, but this is a very active area of research, and further comparison to other methods would be worthwhile.  The method proposed of transferring the representation is well motivated, cleanly described, and conceptually sound.  The assumption that time alignment can be used for the state pairing seems problematic, and should be further validated.

[Official Review · AnonReviewer4 · rating 7 · confidence 3 · 17 Dec 2016 (modified: 20 Jan 2017)]

This paper presents an approach for skills transfer from one task to another in a control setting (trained by RL) by forcing the embeddings learned on two different tasks to be close (L2 penalty). The experiments are conducted in MuJoCo, with a set of experiments being from the state of the joints/links (5.2/5.3) and a set of experiments on the pixels (5.4). They exhibit transfer from arms with different number of links, and from a torque-driven arm to a tendon-driven arm.

One limitation of the paper is that the authors suppose that time alignment is trivial, because the tasks are all episodic and in the same domain. Time alignment is one form of domain adaptation / transfer that is not dealt with in the paper, that could be dealt with through subsampling, dynamic time warping, or learning a matching function (e.g. neural network).

General remarks: The approach is compared to CCA, which is a relevant baseline. However, as the paper is purely experimental, another baseline (worse than CCA) would be to just have the random projections for "f" and "g" (the embedding functions on the two domains), to check that the bad performance of the "no transfer" version of the model is due to over-specialisation of these embeddings. I would also add (for information) that the problem of learning invariant feature spaces is also linked to metric learning (e.g. [Xing et al. 2002]). More generally, no parallel is drawn with multi-task learning in ML. In the case of knowledge transfer (4.1.1), it may make sense to anneal \alpha.

The experiments feel a bit rushed. In particular, the performance of the baseline being always 0 (no transfer at all) is uninformative, at least a much bigger sample budget should be tested. Also, why does Figure 7.b contain no "CCA" nor "direct mapping" results? Another concern that I have with the experiments: (if/how) did the author control for the fact that the embeddings were trained with more iterations in the case of doing transfer?

Overall, the study of transfer is most welcomed in RL. The experiments in this paper are interesting enough for publication, but the paper could have been more thorough.

[Official Review · AnonReviewer3 · rating 6 · confidence 5 · 17 Dec 2016]
**Transfer learning in RL using a nonlinear CCA like approach**

This paper explores transfer in reinforcement learning between agents that may be morphologically distinct. The key idea is for the source and target agent to have learned a shared skill, and then to use this to construct abstract feature spaces to enable the transfer of a new unshared skill in the source agent to the target agent. The paper is related to much other work on transfer that uses shared latent spaces, such as CCA and its variants, including manifold alignment and kernel CCA. 


The paper reports on experiments using a simple physics simulator between robot arms consisting of three vs. four links. For comparison, a simple CCA based approach is shown, although it would have been preferable to see comparisons for something more current and up to date, such as manifold alignment or kernel CCA. A three layer neural net is used to construct the latent feature spaces. 

The problem of transfer in RL is extremely important, and receives less attention than it should. This work uses an interesting hypothesis of trying to construct transfer based on shared skills between source and target agent. This is a promising approach. However, the comparisons to related approaches is not very up to date, and the domains are fairly simplistic. There is little by way of theoretical development of the ideas using MDP theory.

[Public Comment · Coline Manon Devin · 10 Jan 2017]
**Summary of Updates (Jan 10)**

We thank the reviewers for their comments. Most comments asked for additional comparisons and learned state correspondences. 
In response to reviewer comments, we have added the following to our paper:

1. Learning state correspondences by using an alternating optimization to jointly assign the state pairs in P and learn the embedding functions f and g. This significantly relaxes the assumption that states can be paired by time-step and provides better performance in experiment 1. See section 3.3.2 for a description of this method and figures 5 and 7a for results.

2. All of the comparisons suggested by the reviewers: kernel-CCA, unsupervised manifold alignment (Ammar et al. 2015), and random projections. Results are in Figures 5 and 7a.

3. Connections to metric learning and multitask learning in Section 2.

The comparisons have been run on experiments 1 and 2 and will be added to experiment 3 for the final version.

[Public Comment · Coline Manon Devin · 17 Jan 2017]
**Note to reviewers**

Dear reviewers,

We have addressed the concerns presented in the reviews, including comparisons to several prior methods and a method for automatically determining alignment, as detailed in the individual reviewer responses. If you have suggestions for additional comparisons, we would be happy to add them in the final version. As such, we hope you will consider adjusting your reviews accordingly. 

Regards
Abhishek, Coline, Yuxuan, Pieter, Sergey

[Public Comment · Coline Manon Devin · 20 Jan 2017]
**Note to AC and Reviewers**

Dear Reviewers and Area Chair,
The reviewers for the paper provided a positive evaluation, with suggestions for additional experiments providing comparisons with several prior methods, some clarifications, and brought up the important problem of addressing time-based correspondence alignment. We have edited the paper to include additional experiments that address these issues, and therefore believe that the reviewer concerns about the work have been addressed. We discuss the specifics below.

AnonReviewer 1 Comments: 
“Compared to previous work (Ammar et al. 2015)”
- Performed this comparison and added the results into the Figures 5 and 8. We found that our method performed significantly better than this prior work. 
“Is it possible to use dynamic time warping (or similar method) to achieve reasonable results?  Robustness to misspecification of the pairing correspondence P seems a major concern.”
- We introduced an EM style algorithm which removes this assumption. Description of this method are in Section 3.3.2, and results in Figures 5 and 8. 
“more comparison to other transfer methods, including those listed in Sec.2, would be very valuable”
- We have implemented several more baselines: KCCA, Unsupervised Manifold Alignment as suggested by Bou Ammar et al, Direct mappings, random projections, CCA in Figures 5 and 8.

AnonReviewer3 Comments: 
“preferable to see comparisons for something more current and up to date, such as manifold alignment or kernel CCA”, “comparisons to related approaches is not very up to date” 
-Performed comparisons to Kernel CCA and Manifold Alignment using Unsupervised Manifold Alignment (Wang and Mahadevan, Bou Ammar et al) and added to our experiments in Figures 5 and 8. 

AnonReviewer4 Comments: 
“A limitation of the paper is that the authors suppose that time alignment is trivial [...] could be dealt with through subsampling, dynamic time warping, or learning a matching function” 
-We introduce a new EM style algorithm alternating between feature learning and dynamic time warping. Description is in Section 3.3.2 and results in Figures 5 and 8. 
“Another baseline (worse than CCA) would be to just have the random projections for "f" and "g"”
-We added this comparison in Figures 5 and 8.
“at least a much bigger sample budget should be tested [...] control for the fact that the embeddings were trained with more iterations in the case of doing transfer” 
- We ran the baseline with a significantly higher sample budget as shown in Table 1. The poor performance is likely due to not enough guided exploration happening without good reward shaping for the baseline. 
“problem of learning invariant feature spaces is also linked to metric learning [...] no parallel is drawn with Multi-Task learning in ML ”
-The additional references suggested have been added in the Related Work (Section 2).

[Final Decision · Program Chairs · 06 Feb 2017]
**ICLR committee final decision**

pros:
 - tackles a fundamental problem of interest to many
 - novel approach
 
 cons:
 - originally not evaluated against some reasonable benchmarks. Note: now added or addressed
 - little theoretical development cf MDP theory
 - some remaining questions about the necessity (and ability) to find good time alignments
 
 I personally found the ideas to be quite compelling, and believe that this is likely to inspire future work.
 The experiments represent interesting scenarios for transfer, with the caveat that they are just in simulation.